# Direct Visualization of Amlodipine Intervention into Living Cells by Means of Fluorescence Microscopy

**DOI:** 10.3390/molecules26102997

**Published:** 2021-05-18

**Authors:** Christine Quentin, Rūta Gerasimaitė, Alexandra Freidzon, Levon S. Atabekyan, Gražvydas Lukinavičius, Vladimir N. Belov, Gyuzel Y. Mitronova

**Affiliations:** 1Department of NanoBiophotonics, Max Planck Institute for Biophysical Chemistry, Am Fassberg 11, 37077 Göttingen, Germany; christine.quentin@mpibpc.mpg.de (C.Q.); vladimir.belov@mpibpc.mpg.de (V.N.B.); 2Chromatin Imaging and Labeling Group, Department of NanoBiophotonics, Max Planck Institute for Biophysical Chemistry, Am Fassberg 11, 37077 Göttingen, Germany; ruta.gerasimaite@mpibpc.mpg.de (R.G.); grazvydas.lukinavicius@mpibpc.mpg.de (G.L.); 3Federal Research Center Crystallography and Photonics, Photochemistry Center, Russian Academy of Sciences, Novatorov 7a, 119421 Moscow, Russia; freidzon.sanya@gmail.com (A.F.); levat51@mail.ru (L.S.A.); 4National Research Nuclear University MEPhI (Moscow Engineering Physics Institute), Kashirskoye Shosse 31, 115409 Moscow, Russia

**Keywords:** amlodipine, fluorescence, live cell imaging, microscopy, lysosomes

## Abstract

Amlodipine, a unique long-lasting calcium channel antagonist and antihypertensive drug, has weak fluorescence in aqueous solutions. In the current paper, we show that direct visualization of amlodipine in live cells is possible due to the enhanced emission in cellular environment. We examined the impact of pH, polarity and viscosity of the environment as well as protein binding on the spectral properties of amlodipine in vitro, and used quantum chemical calculations for assessing the mechanism of fluorescence quenching in aqueous solutions. The confocal fluorescence microscopy shows that the drug readily penetrates the plasma membrane and accumulates in the intracellular vesicles. Visible emission and photostability of amlodipine allow confocal time-lapse imaging and the drug uptake monitoring.

## 1. Introduction

Amlodipine (AML) belongs to a 1,4-dihydropyridine (DHP) family of L-Type Ca^2+^-channels (LCC) blockers [1,2,3,4,5]. The drug is widely used to treat high blood pressure, coronary artery disease, and angina pectoris [3]. Recent data show that AML has potent anti-SARS-CoV-2 activity [6,7]. Its physicochemical, pharmacological and pharmacokinetic properties differ from those of other LCC blockers in DHP group, mainly due to the presence of 2-(2-aminoethoxy)methyl side group in the 1,4-dihydropyridine residue (Figure 1a) [1,5]. One of these properties is very slow rate of AML interaction with the Ca_V_1.2α, a membrane subunit of LCC that forms a Ca^2+^-selective pore and regulates the Ca^2+^ influx into the cell [1,2,8]. The long-lasting Ca^2+^ antagonistic activity of AML is attributed to its slow on/off- kinetics and to its electrostatic interactions with the phospholipid head groups leading to drug concentration in the membrane [8]. AML has weak fluorescence in aqueous solutions (λ_ex_ = 365 nm; λ_em_ = 450 nm; *Φ* ~ 0.04); but shows up to 9-fold fluorescence enhancement in human serum albumin (HSA) and human plasma [9,10].

Our calculations show that the fluorescence loss in aqueous solutions is caused by the nonradiative relaxation and proceeds through the conical intersection between the ground state (S_0_) and the lowest excited singlet state (S_1_). This is achieved by the rotation of 2-chlorophenyl ring followed by the DHP fragment deformation. Suppression of such large amplitude motion in AML, for example by a viscous medium, leads to the fluorescence enhancement. When bound to a protein, the motion of the ligand is severely limited [11]. If the binding event is accompanied with the emission increase, it becomes easily detectable. Using fluorescence saturation assay, we were able to measure the binding affinity of AML to Bovine Serum Albumin (BSA) and HSA. Though the fluorescence of AML has been previously reported [9,10], no attempt has been made so far to use its emission for visualization of the drug distribution in living cells. We found that the AML fluorescence is sufficiently bright and stable to allow its visualization inside living cells by confocal microscopy. We demonstrated that AML is rapidly sequestered into acidic compartments—late endosomes and lysosomes, and propose that these compartments serve as drug pools, which contribute to the long-lasting activity of AML.

## 2. Results

Upon AML binding to the BSA/HSA hydrophobic pocket, the fluorescence intensity (*I_fl_*) increases (Appendix A) [9]. We observed that the quantum yield (*Φ*) and the lifetime (*τ*) of AML fluorescence were significantly higher in the presence of BSA or HSA than for a free drug (Table 1). The self-aggregation of AML is not responsible for this emission enhancement, as addition of anionic surfactant, sodium dodecyl sulfate (SDS), had little effect on *I_fl_* of AML (Appendix A). To study whether the ligand translocation from the polar solvent to nonpolar binding site can lead to fluorescence enhancement, we recorded AML absorption and emission spectra in different solvents and found no correlation between dielectric constant and fluorescence increase. In general, the *I_fl_* enhancement, compared with aqueous AML solutions, was stronger in polar, aprotic solvents and glycerol (Appendix A). The bathofluoric shift (~80 nm) in DMSO, DMF and DMAA solutions is presumably related to an intermolecular N-H…O hydrogen bond between AML and a solvent [13]. The double-peak emission band (430 and 445 nm) observed in BSA/HSA solutions may be a vibronic effect, which is supported by quantum chemical calculations on the model compound, dimethyl 4-(2-chlorophenyl)-2,6-dimethyl-1,4-dihydropyridine-3,5-dicarboxylate (Appendix A).

Fluorescence enhancement in glycerol may be due to high viscosity, which leads to a slower deactivation through non-radiative processes [14,15]. To test this, we performed fluorescence measurements of AML in methanol/glycerol mixtures of different viscosities (*η*); their dielectric constants 32.6 and 42.5 respectively (Figure 1b). *I_fl_* and *Φ* increased dramatically with increasing *η* (Appendix A). The calculated radiative (*k_0_*) and non-radiative (*k_nr_*) decay constants illustrate this effect with *k_0_* remaining almost constant and decreasing *k_nr_* with increasing viscosity of the medium (Appendix A). Thus, partial constraining of AML molecular twisting and rotations by a viscous medium prevents its transition to a non-emissive “dark” state. In a similar fashion, the specific interactions between the ligand and the protein’s backbone may restrict molecular flexibility of AML and enhance its fluorescence in BSA/HSA solutions.

To assess the mechanism of the fluorescence quenching in aqueous solutions we performed quantum-chemical calculations of AML in the ground and exited states. In the ground state (S_0_) the calculated molecular structure of AML is similar to the one obtained by single crystal X-ray analysis [12]. The DHP and 2-chlorophenyl moieties almost perpendicular to each other due to the presence of the bulky ester groups and the DHP fragment has a boat conformation (Figure 1a). After absorption of the light, the exited molecule relaxes to the lowest excited singlet state S_1_, from which it emits a photon and the fluorescence occurs. The molecular geometry of AML in S_1_ does not change significantly from the one in the S_0_ (Appendix A). The dissipation of energy responsible for the fluorescence loss in aqueous solutions can occur because of the 2-chlorophenyl group rotation. However, our computations showed that the rotation of 2-chlorophenyl group alone could not lead to the fluorescence quenching. We found a conical intersection between the lowest excited singlet state S_1_ and S_0_ separated from S_1_ by the 6.1 kcal/mol barrier. With such barrier and the calculated frequency of the reorganizational mode 61 cm^−1^, the molecule may achieve the conical intersection point in ~15 ns. Figure 2 depicts the molecular conformations and potential energy surface of AML and the structures of the transition state and the conical intersection of AML. The structure of the conical intersection involves 2-chlorophenyl ring twist with simultaneous DHP ring puckering (Figure 2b). This conical intersection is only accessible through the large-amplitude motion, which is hindered in viscous media. In non-viscous media, the barrier can be overcome at a rate comparable with the radiative relaxation channel, which leads to fluorescence quenching. The calculated radiative *τ_rad_* for AML is ~22 ns, which is in rather good agreement with the measured values in glycerol, 14.0 ± 0.2 ns, and in 10% HSA, 16.8 ± 0.3 ns (Table 1).

We performed laser-flash photolysis measurements to test whether atmospheric oxygen can change the spectrum or kinetics of the photo-induced absorption of AML. The measurements indicate the presence of intersystem crossing to the triplet state in both PBS and PBS-containing 10% BSA (Appendix A). We have not detected any phosphorescence of AML in these solutions at room temperature. The presence of a non-emissive triplet state of AML in 10% BSA solution is evidenced by the fact that its delayed fluorescence spectrum measured in the phosphorescence mode (delay 200 μs) is similar to its prompt fluorescence spectrum.

In order to ascertain whether AML fluorescence is sensitive to intramolecular charge transfer between DHP and aryl moieties, we synthesized the 2-chloro-5-carboxy AML derivative (AML-5-COOH, for the synthesis see Appendix A), which differs from AML in an additional carboxyl group attached to the 2-chlorophenyl moiety. In contrast to AML, AML-5-COOH does not fluoresce. The calculated absorption and emission spectra (Appendix A) agree with the experimental results: the oscillator strengths (f) of both absorption and emission of AML are 2–5 times higher than those of AML-5-COOH. The quantum-chemical calculations showed that both electron density of the highest occupied (HOMO) and lowest unoccupied (LUMO) molecular orbitals of AML are localized mainly on DHP, while in AML-5-COOH, containing the electron-withdrawing substituent COOH, LUMO is localized on both DHP and aromatic ring (Appendix A). This fact indirectly confirms that the light absorption by AML does not lead to the intramolecular charge transfer, but, taking into account the delayed fluorescence, includes the intersystem crossing into a DHP localized triplet state.

The enhancement of AML fluorescence upon binding to BSA or HSA allowed us to determine the AML-protein binding affinities by fluorescence titration experiments. For that, we recorded the fluorescence spectra of 5 µM AML or (*S*)-AML in the 0–750 µM protein (BSA or HSA) solutions and observed the fluorescence intensity changes at the emission maximum (Appendix A). The fluorescence signal saturated above 200 µM of protein concentration (Figure 1c and Appendix A), and the calculated dissociation constants (*K_d_*) were in a good agreement with previous studies where the tryptophan (Trp) fluorescence quenching was used for the binding constant determinations [9,16]. The measuring of AML emission with longer excitation wavelength (λ), than *λ*_em_ (Trp) = 350 nm helps to minimize autofluorescence.

Next, we tested whether AML fluorescence is suitable for microscopy experiments on live cells and used the A7r5 vascular smooth muscle cells and HL-1 cardiomyocytes. These cells express the main AML target, Ca_V_1.2, in cardiac cells. Time-lapse confocal microscopy showed a drug accumulation in the cytosol as punctate structures associated with cytoplasmic vesicles (Appendix A). The emission maximum measured on cells was equivalent to AML emission in BSA/HSA solutions (Figure 1f). The time-lapse confocal imaging revealed that AML fluorescence is stable; we detected almost no photobleaching after imaging of 18 frames using 2.5 mW laser power; measured power density 17 kW/cm^2^ (Appendix A). However, the labelling pattern was markedly different from the immunostaining with anti Ca_V_1.2α antibody (Appendix A and Reference [17]). In order to elucidate whether internalization of AML depends on the interaction with Ca_V_1.2, we repeated the staining in the presence of non-fluorescent DHP agonist (*S*)-(−)-Bay K8644.12. (*S*)-(−)-Bay K8644 had no effect on cell staining by AML (Figure 3), indicating that AML clusters are not associated with Ca_V_1.2. In addition, we did not observe any differences in AML staining using confocal microscopy on live HEK-293 and HEK-293 Ca_V_1.2 cells, containing inducible genes of human Ca_V_1.2 (Appendix A). Although HEK-293 cells produce some DHP-sensitive Ca^2+^ currents, their magnitude considerably increases upon expression of recombinant Ca_V_1.2α [18]. If interaction with receptors is required for AML internalization, overexpression of Ca_V_1.2 should increase the staining efficiency.

Altogether, our data show that unbound AML accumulates in the intracellular compartments independently of its interaction with Ca_V_1.2 channels, the main DHP target in cardiac cells. There are two possibilities why we could not observe staining of Ca_V_1.2 by AML: either interaction with the channel does not lead to fluorescence increase, presumably because AML takes the molecular conformation with lower fluorescence emission than in a stabilized form in BSA/HSA solutions, or the density of receptors is too low for the detection and is masked by receptor-independent staining.

Importantly, the shape and the size of HEK-293 Ca_V_1.2 cells, before and after Ca_V_1.2 expression induction with doxycycline, are remarkably different. We noticed the enlargement of the cells after the induction; some of them appeared as multinucleated giant cells (Appendix A).

To identify compartments stained with AML, we performed co-localization study with several cellular live-cell stains: MitoTracker^®^ Orange (mitochondria), ER-Tracker^®^ Red (endoplasmic reticulum); and genetic markers: Golgi-emGFP, Rab7a-emGFP (late endosomes), Rab5a-emGFP (early endosomes) and LAMP1-emGFP (lysosomes). We treated living A7r5 cells with the respective marker and AML, and then visualized using a confocal Leica SP8 microscope. After 1 h incubation, the late endosomal and lysosomal labelling showed the co-occurrence of the cyan (AML) and green channel (emGFP) emissions with calculated Pearson’s co-localization coefficients (*R*) above threshold: *R* ~0.4 (Figure 4). Rather modest *R* value can be explained by the fact that Rab7a/LAMP1-emGFP are membrane markers of late endo-/lysosomes, while AML mainly accumulates inside the acidic lumen of these organelles (Figure 4, for Z-stack see Appendix A). AML staining displays no correlation with Golgi or early endosome trackers and only a weak positive correlation with the ER and mitochondria after 1 h incubation with the drug (*R* = 0.07, 0.08, 0.22 and 0.26, respectively, Appendix A). These findings allow us to conclude that AML–associated structures in the perinuclear region are late endocytic organelles.

In order to clarify how AML crosses the membrane, we stained A7r5 cells with the endocytosis tracer FM 4-64 [19]. FM 4-64 is a lipophilic fluorogenic dye with emission in a hydrophobic environment. After 15 min incubation of the cells with AML and FM 4-64, we observed both compounds inside the cells, but they did not co-localize (Figure 5), which suggested different mechanism of their entry. On the contrary, after 75 min of incubation, FM 4-64 and AML co-localized with *R* = 0.61 ± 0.15 (Figure 5 and Appendix A). Due to its positively charged tetraalkyl ammonium group, FM 4-64 does not diffuse through the membranes and thus travels through the endocytic route from early to late endosomes and then to lysosomes [20]. In contrast, the direct AML loading to the A7r5 cells stained with endo-/lysosomal markers revealed that AML accumulates in lysosomes from the very beginning (Appendix A). This strongly indicates that endocytosis is not the main route for AML to enter the cells. This fact goes in line with the lysosomal sequestration mechanism discovered by De Duve et al. [21]. Upon diffusion into acidic compartments, such as late endo-/lysosomes, the equilibrium between charged and uncharged forms of a cationic drug shifts toward the protonated form and its membrane permeability decreases, which facilitates its retention within the compartments [22]. To further test this hypothesis, we checked whether AML accumulation could be reversed by addition of NH_4_Cl, the inhibitor of lysosomal trapping [22]. As expected, the addition of 10 mM NH_4_Cl to the AML treated cells substantially reduced AML staining (Figure 6).

## 3. Discussion

In hydrophobic environments, AML has poor fluorescence and cannot be readily detected, while incorporated into a lipid bilayer. Consistently, we see no fluorescence on the plasma membrane. Upon accumulation in the acidic compartments, the internalized drug is exposed to considerably lower pH, and this can be a reason for fluorescence enhancement. We evaluated the pH sensitivity of AML and found that the fluorescence of AML did not change significantly in the physiological pH range (Appendix A). Because suppression of the large amplitude motion in AML (and other fluorophores) leads to the fluorescence enhancement, the viscosity of cellular compartments can contribute to its visibility. In the highly heterogeneous environment of a live cell, the microviscosity in cell vesicles can reach 140 cP, while in the cellular cytoplasm it is less than 35 cP [23,24].

Figure 7 depicts a simplified model of AML translocation into the cytosol and its lysosomal sequestration. AML molecules in an extracellular region exist in both non-protonated and protonated forms. Under physiological conditions, the cytosolic pH is only slightly acidic (7.0–7.2) compared to the extracellular pH (7.2–7.4). Upon AML internalization into a cytosol, its non-protonated fraction remains almost the same according to the Henderson–Hasselbalch relationship [22]. In a phospholipid membrane, AML molecules are non-fluorescent, at least until they are immobilized in a specific configuration favoring the fluorescence emission. In the concentrations used in our study, AML acts as a long-lasting Ca^2+^ channel blocker (Appendix A), but apparently locates not exclusively to its target, Ca_V_1.2. The substance penetrates rapidly into a cytosol where it can be detected as a dim staining; already in the first 3–5 min of the fluorescence readout. After diffusion into the late endo-/lysosomal lumen, the drug gets protonated and trapped inside. The elevated viscosity of the acidic compartments suppresses the high amplitude movements of excited AML molecules, and the emission of light occurs. Upon accumulation of the drug in the late endo-/lysosomal compartments, the fluorescence increases indicating high concentrative capacity of the acidic vesicles. The rapid entry of AML and its intracellular distribution, observed in live cells by means of optical microscopy, allow us to assume that the acidic compartments might serve as an additional drug pool. Because the small amount of uncharged AML molecules is always present, they may escape the lysosomes and reach their target Ca_V_1.2 on the plasma membrane.

One can speculate that AML fluorescence might appear as a result of lysosomal enzymatic activities, but, contrary to that, the primary pathway of AML metabolism in humans and animals is an oxidation to the non-fluorescent pyridine analogs [25,26]. The metabolites with DHP structures as the products of the ester cleavage or oxidative deamination of the 2-aminoethoxymethyl side chain were found in rats and dogs. These metabolites as zwitterionic or negatively charged molecules would not interfere with negatively charged intralysosomal vesicles responsible for the lysosomal trapping mechanism, and most probably will be released into the cytosol. Remarkably, no AML metabolites show any Ca^2+^ channel blocking activities [25,27].

In summary, we examined the fluorescence properties of the well-known antihypertensive drug amlodipine in the viscous media and upon binding to transport proteins. We demonstrated that amlodipine fluorescence enhancement in cellular environment can be used for its visualization inside living cells by confocal microscopy. A weakly basic amlodipine rapidly enters the cell and accumulates inside Rab7a-positive late endosomes and LAMP1-positive lysosomes, in its “stabilized” form, that favors its fluorescence. The molecule of the drug achieves this “stabilized” form by suppression of the 2-chlorophenyl ring rotation followed by the DHP fragment deformation. We propose that accumulation of amlodipine in the acidic compartments might contribute to its slow onset and long-lasting action.

## 4. Materials and Methods

### 4.1. General Materials and Methods

Racemic amlodipine besylate (97+%) was purchased from Alfa Aesar (Kandel, Germany). (*S*)-(–)-amlodipine ((*S*)-AML) besylate (98 +%) was purchased from Carbosynth (Berkshire, UK). (*S*)-(−)-Bay K8644 was purchased from Tocris Bioscience (Wiesbaden-Nordenstadt, Germany). BSA (98 +%), HSA (Fraction V) and Phosphate Buffered Saline tablets for 0.01 M phosphate buffer (PBS) were purchased from Merck, Darmstadt, Germany. All reagents were used without further purification. To prepare 20 mL of 10% BSA/HSA solutions in phosphate buffered saline (PBS, pH 7.4), 2 g of protein was dissolved in a 20 mL volumetric flask with the previously prepared PBS. The protein solution was kept in a refrigerator until use.

Fluorescence emission and absorption spectra were recorded on a multiwell plate reader TECAN Spark 20M in 96-well glass bottom plates (MatTek Corporation; Cat. No. PBK96G-1.5-5-F) at room temperature (25 °C). Fluorescence emission of AML was recorded from 390 nm to 600 nm with excitation at 360 nm. The emission bandwidth was set to 10 nm, excitation bandwidth—7.5 or 10 nm, gain—100, number of flashes—20. The absorption spectra were recorded: from 300 to 430 nm. The background absorption of a glass bottom plate was measured in wells containing only buffer and subtracted from the spectra of the samples. The spectra were averaged from three individual experiments. The absorption spectra of AML in glycerol/methanol mixtures were recorded on a Varian Cary 4000 UV-Vis spectrophotometer in quartz cuvettes (3 mL) with a 1 cm path length. Fluorescence spectra of 5 µM AML in PBS solutions of different pH (3–9) were recorded on a Varian Cary Eclipse fluorescence spectrometer.

Absolute fluorescence quantum yields (*Φ*) were obtained on a Quantaurus-QY absolute PL quantum yield spectrometer (model C11347-12, Hamamatsu Hamamatsu Photonics Deutschland GmbH, Herrsching am Ammersee, Germany) at ambient temperature (25 °C), excitation wavelength 370 nm; all measurements were performed in triplicates. Fluorescence lifetimes (*τ*) were measured with a Quantaurus-Tau fluorescence lifetime spectrometer (model C11367-32, Hamamatsu, Hamamatsu Photonics Deutschland GmbH, Herrsching am Ammersee, Germany). All measurements were performed in air-saturated solvents at ambient temperature. Radiative fluorescence lifetime (Table 1, *τ_rad_*) was calculated by the formula: *τ_rad_* = *τ*/*Φ*.

### 4.2. Laser-Flash Photolysis Experiments

LFP experiments were performed using a Q-switched Nd:YAG laser (355 nm, 70 mJ per pulse, 10 ns half width, SOLAR laser system, Minsk, Republic of Belarus). All transient spectra were recorded using 10 × 10 mm quartz cells with a capacity of 4 mL and were bubbled for 30 min with argon before acquisition. All the experiments were conducted at room temperature.

### 4.3. Quantum Chemical Calculations

The molecular geometry of the compound and its spectra were calculated using the FireFly program package [28] partially based on GAMESS [29]. The geometry was optimized by the density functional theory (PBE0/6-31+G(d,p)); the spectra, excited-state geometry, transition states and conical intersection points were calculated by the time dependent density functional theory with the same functional and basis set.

The radiative rate constant was calculated by the formula: $k_{0}=\frac{2}{3}\nu^{2}f$, where ν is the transition frequency in cm^−1^ and *f* is its oscillator strength. The rate of achieving the conical intersection via the barrier was calculated by the Arrhenius formula: $k_{nr}=\omega e^{-A/k_{b}T}$, where ω is the frequency of the reorganizational mode in s^−1^ and *A* is the barrier height.

### 4.4. Determination of the Apparent Dissociation Constants (K_d_) by Fluorescence Enhancement

The tested compounds were dissolved in the PBS containing different concentrations of BSA or HSA (0–10% in PBS, pH 7.4 + 0.1 *v*/*v*% DMSO). The obtained solutions were incubated for 1 h at room temperature to ensure ligand binding to the proteins. Emission spectra of BSA and HSA solutions, as well as the spectra of BSA/HSA solutions containing AML or (*S*)-AML (5 µM), were measured on a multiwell plate reader as given in “General materials and methods” section; the emission wavelength step size was 1–2 nm. The fluorescence intensities at 450 nm were measured for the calculation of the AML fluorescence increase upon binding to BSA/HSA. The fluorescence increase of compounds in BSA or HSA in relative fluorescence units (RFU) was plotted vs. proteins concentrations. Binding constants were calculated using the software package GraphPad Prism version 8.3.1 (GraphPad Software, San Diego, CA, USA). Y = (B_max_ × X/(*K_d_* + X)) + off was used, where B_max_ is the maximum specific binding.

### 4.5. Cell Culture

HEK-293 Ca_V_1.2 cells with stable expression of human Ca_V_1.2 calcium channels (B’SYS, Witterswil, Switzerland) were cultured in DMEM/F-12 (Dulbecco’s Modified Eagle medium, Thermo Fisher Scientific, Darmstadt, Germany) supplemented with GlutaMAX-1 (Thermo Fisher Scientific), 10% fetal bovine serum (FBS) (Thermo Fisher Scientific, Darmstadt, Germany) and 0.9% Penicillin/Streptomycin (Merck, Darmstadt, Germany) in a humidified 5% CO_2_ incubator at 37 °C. In addition, we used a reduced antibiotic pressure for the cultivation of the cells; 100 µg/mL hygromycin B (Thermo Fisher Scientific, Invitrogen), 15 µg/mL Blasticidin (Invivogen, Toulouse, France), 0.4 µg/mL puromycin (Invivogen, Toulouse, France). To induce the expression of Ca_V_1.2, 0.1 µg/mL doxycycline (Merck, Darmstadt, Germany) was added 24 h before experiments.

A7r5 cells (ATCC, Wesel, Germany) were cultured in DMEM Medium supplemented with 10% FBS, 0.9% penicillin/streptomycin in a humidified 5% CO_2_ atmosphere at 37 °C. Cells were split every 3–5 days or at confluence.

HL-1 cells (Merck Millipore, SCC065) were cultivated in Claycomb’s Medium (Sigma), supplemented with Glutamax, 10% fetal bovine serum, 0.9% penicillin/streptomycin and 0.1 mM noradrenaline (Sigma) on Fibronectin/Gelatine (Sigma) coated culture bottles or plates.

### 4.6. Ca^2+^ Assay

For fluorescent plate reader Ca^2+^ assay, HEK-293 Ca_V_1.2 cells (20,000 per well) or HL-1 cells (25,000 per well) were cultivated for 24 h in black-walled, clear-bottom 96-well microplates (Corning, Amsterdam, The Netherlands) covered with fibronectin (10 μg/mL, Roche, Mannheim, Germany). Cells were grown in a humidified incubator at 37 °C and 5% CO_2_.

Changes in Ca^2+^ concentration were measured on a multiwell plate reader TECAN Spark 20M using FLIPR 6 Calcium Assay Kit from Molecular Devices (Molecular Devices LLC, München, Germany) at room temperature (25 °C). The 1000× stock solutions of AML were prepared in DMSO. Briefly, the culture medium was removed and the 100 µL loading buffer was added to the cells together with 100 µL AML solutions of different concentrations in Hanks’ balanced salt solution (HBSS, Lonza) containing 20 mM 4-(2-hydroxyethyl)-1-piperazineethanesulfonic acid (HEPES). Then the cells were incubated for 2 h at 37 °C. The cells were washed 2 times with 200 µL HBSS containing AML or (*S*)-AML (1 nM–5 µM) and 20 mM HEPES. The time courses were recorded in each well. Fluorescence was evoked by 485 nm excitation wavelength and collected in a bottom-read mode at 525 nm. Data was recorded every 2 s (exposure—20 flashes, excitation—10 nm, emission bandwidth—15 nm). To initiate Ca^2+^ influx, 50 µL of 400 mM KCl solution in HBSS was injected in each well (speed 100 µL/s) at 10 s time point. To get a fluorescence maximum, 15 of 30 µM Ionomycin (Merck, Darmstadt, Germany) solution in HBSS was injected in each well (speed 100 µL/s) at 50 s time point. The ratios between the KCl induced peak minus basal fluorescence, ΔF(KCl), were taken and expressed as a quotient of the maximal increase induced by ionomycin, ΔF (Ionomycin). The dose response curves of AML were calculated using the software package GraphPad Prism version 8.3.1. All data are presented as mean ± SD in 4 independent experiments. Y = Bottom + (Top-Bottom)/(1 + 10^((LogIC_50_ − X) × HillSlope)) was used where HillSlope describes the steepness of the curve, Top and Bottom are plateaus in the units of the Y axis.

### 4.7. Cells Preparation for Live-Cell-Imaging Experiments

For the wide-field microscopy experiments cells were seeded in µ-24 well plates (Ibidi, Gräfelfing, Germany) or glass bottom 12-well plates (MatTek Corporation, Bratislava, Slovak Republic) at 25,000 (A7r5 and HL-1) or 12,500 (HEK-293 Ca_V_1.2) cells per well. For the confocal imaging cells were seeded in CELLview™ Slides (Greiner Bio-One, Frickenhausen, Germany); 8000 cells per well or in 35 mm glass bottom dish (Cellvis, USA); well 14 mm, glass thickness #1.5, 30,000 cells per well. The induction of HEK-293 Ca_V_1.2 cells was done by 24 h incubation in doxycycline solution (0.1 µg/mL). For cellular imaging, cells were incubated at 37 °C with 0.1–2 µM AML prepared in Hanks’ balanced salt solution (HBSS) supplemented with 20 mM 4-(2-hydroxyethyl)-1-piperazineethanesulfonic acid (HEPES) pH 7.4, from a 10 mM stock solution prepared in DMSO. For co-staining experiments cell were labeled with CellLight^®^ Golgi-emGFP (fusion construct of Human Golgi-resident enzyme *N*-acetylgalactosaminyltransferase 2 and emGFP), MitoTracker^®^ Orange, ER-Tracker^®^ Red, CellLight^®^ Early Endosomes-emGFP (fusion construct of Rab5a and emGFP), CellLight^®^ Late Endosomes-emGFP (fusion construct of Rab7a and emGFP), BacMam 2.0 (ThermoFisher Scientific, Schwerte, Germany) according to the manufactures instructions. 5 µM SynaptoRed C2^®^ (FM 4-64, Tocris, Wiesbaden-Nordenstadt, Germany) was used for membrane staining. For the nucleus staining, we used 1 µM TMR-Hoechst and 100 nM *610CP*-5-COOH-Hoechst probe [30]. Samples were washed two times with HBSS/HEPES, and imaged in the same buffer.

### 4.8. Fluorescence Microscopy

Confocal imaging was performed on a Leica SP8 (Leica Microsystems, Mannheim, Germany) inverted confocal microscope equipped with an HC PL APO CS2 63x/1.40 Oil objective. All live-cell images shown in the main text were acquired using a 700 Hz bidirectional scanner, and a 92.26 × 92.26-μm field of view (1024 × 1024 pixels and 90 nm pixels), pinhole of 95.6 µm diameter (1 AU), pixel dwell time—325 ns. AML was excited with 405 nm laser (3% intensity) and detected with a regular PMT in the 420–470 nm range. CellLight^®^ Golgi-emGFP, CellLight^®^ EarlyEndosomes-emGFP and Late Endosomes-emGFP (fusion construct of Rab7a and emGFP) were excited with a 488 nm laser (5% intensity) and detected with a regular PMT in the 510–560 nm range. ER-Tracker^®^ Red was excited with a 561 nm laser and detected with the Leica HyD detector set within the spectral range of 600–650 nm. SynaptoRed C2 (FM 4-64) was exited with a 405  or 561 nm laser and detected with the Leica HyD detector set within the spectral range of 590–650 nm. *610CP*-5-COOH-Hoechst probe was excited with 610 nm laser and detected with a regular PMT in the 635–650 nm range. For wide-field imaging we used Lionheart FX automated microscope (Biotek) equipped with dry air objectives 4×, 20× and 40× and sample injection system. 16 fields of view in 3 focusing planes, spanning 6 μm in thickness, were acquired per well. Image stitching and focus stacking was performed using in-built Gene 5 software (Biotek). The final images encompassed 1380 × 950 μm field of view and contained at least 220 and up to 2000 cells.

The images in Figure 4 were deconvolved with Huygens Essential version 20.10 (Scientific Volume Imaging, The Netherlands, http://svi.nl, accessed on 19 February 2021). All acquired or reconstructed images were processed and visualized using Huygens Essential 20.10 or ImageJ 1.53c software [31]. Co-localization analysis was carried out on whole images (Huygens Essential 20.10) or on five to ten randomly selected regions of interest (ROIs, ImageJ 1.53c). Co-localization was established for pixels whose intensities were higher than threshold.

## Figures and Tables

**Figure 1 molecules-26-02997-f001:**
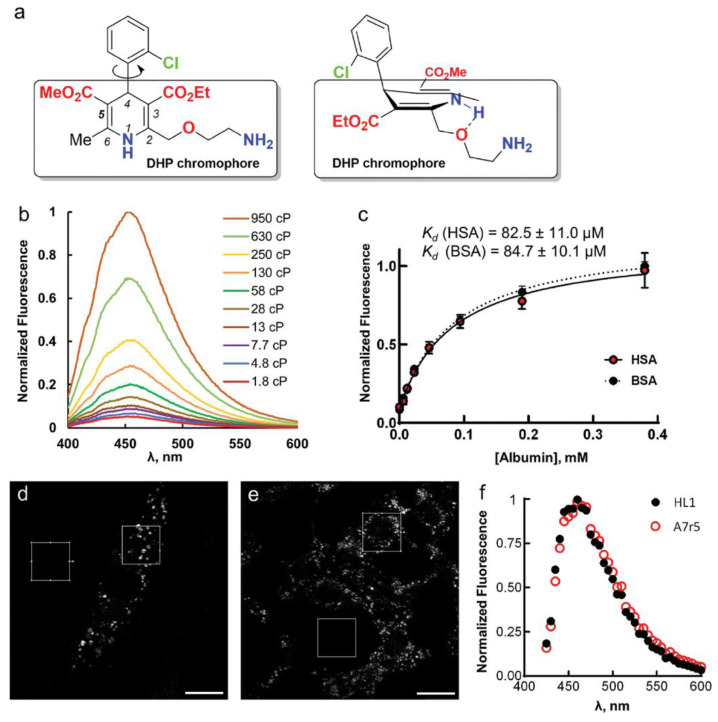
(**a**) 2-((2-Aminoethoxy)methyl)-4-(2-chlorophenyl)-3-ethoxycarbonyl-5-methoxycarbonyl-6-methyl-1,4-dihydropyridine, amlodipine. Planar structure and molecular geometry according to the crystallographic data are shown [12]. DHP chromophore is framed. The dashed line shows a hydrogen bond. (**b**) Fluorescence spectra of 5 µM AML in MeOH/glycerol mixtures of different viscosity. (**c**) Equilibrium binding of (*S*)-AML to BSA and HSA measured by titrating 5 μM (*S*)-AML with 6 µM to 0.38 mM BSA/HSA. Values are mean ± S.D., *n* = 3 independent measurements. (**d**,**e**) Confocal images of living A7r5 (**d**) and HL-1 (**e**) cells stained with 5 µM AML obtained with 405 nm excitation laser. Scale bars—15 µm. To obtain the emission spectrum, the mean signal was measured in a selected region of interest (15 × 15 µm^2^) and in an equivalent region without the cells. The spectra from three regions of interest (ROIs) were averaged and the background was subtracted to yield spectra shown in (**f**).

**Figure 2 molecules-26-02997-f002:**
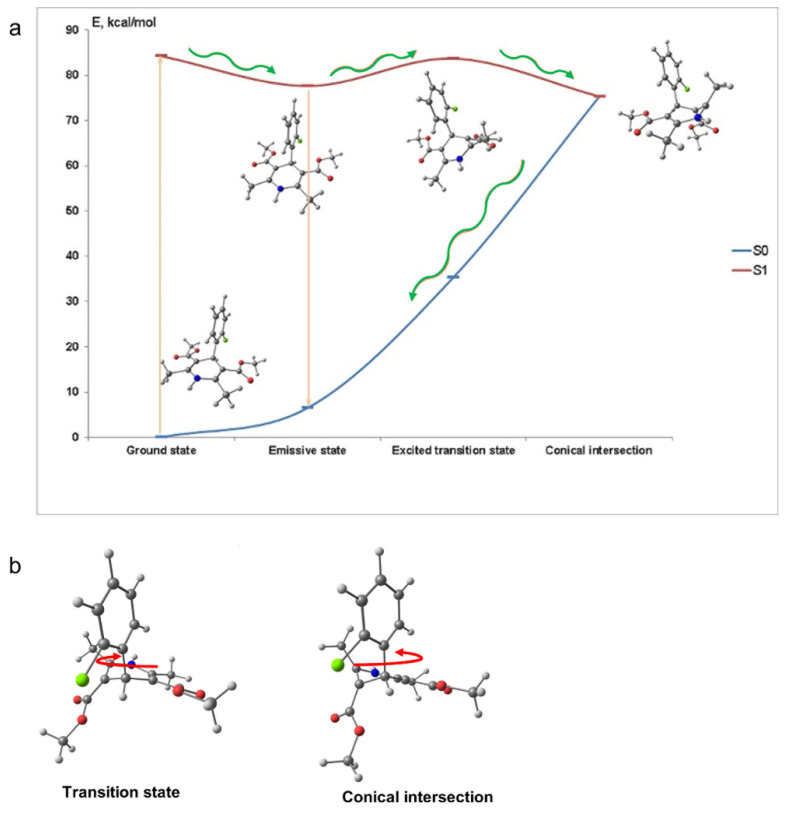
(**a**) Molecular conformations and potential energy surface scan of AML. The exited molecule (vertical arrow up) relaxes to the minimum of the S_1_ state (wavy arrow), from which it can emit a photon (vertical arrow down). When the drug is in the viscous or crowded environment, further rotation of the 2-chlorophenyl ring is hindered, and only fluorescence is observed. In aqueous solution, the molecule can overcome the barrier from the S_1_ minimum (green wavy arrow) and go to the S_1_-S_0_ conical intersection (2.3 kcal/mol below the S_1_ minimum, green wavy arrow), from which it will relax to the ground state non-radiatively (green wavy arrow). This leads to the fluorescence quenching. (**b**) The structures of the transition state and the conical intersection. 2-chlorlphenyl ring twist with simultaneous DHP ring puckering is shown (C-COOR group goes out of plane pertinent to the double >C=C< bond with adjacent carbon atoms). Red arrows indicate the direction of the 2-chlorophenyl ring twist to achieve the conversion from the transition state to the conical intersection and from the conical intersection to the ground state. The atomic coordinates of the model AML in the S_0_, S_1_, transition states and conical intersection are listed in Appendix A.

**Figure 3 molecules-26-02997-f003:**
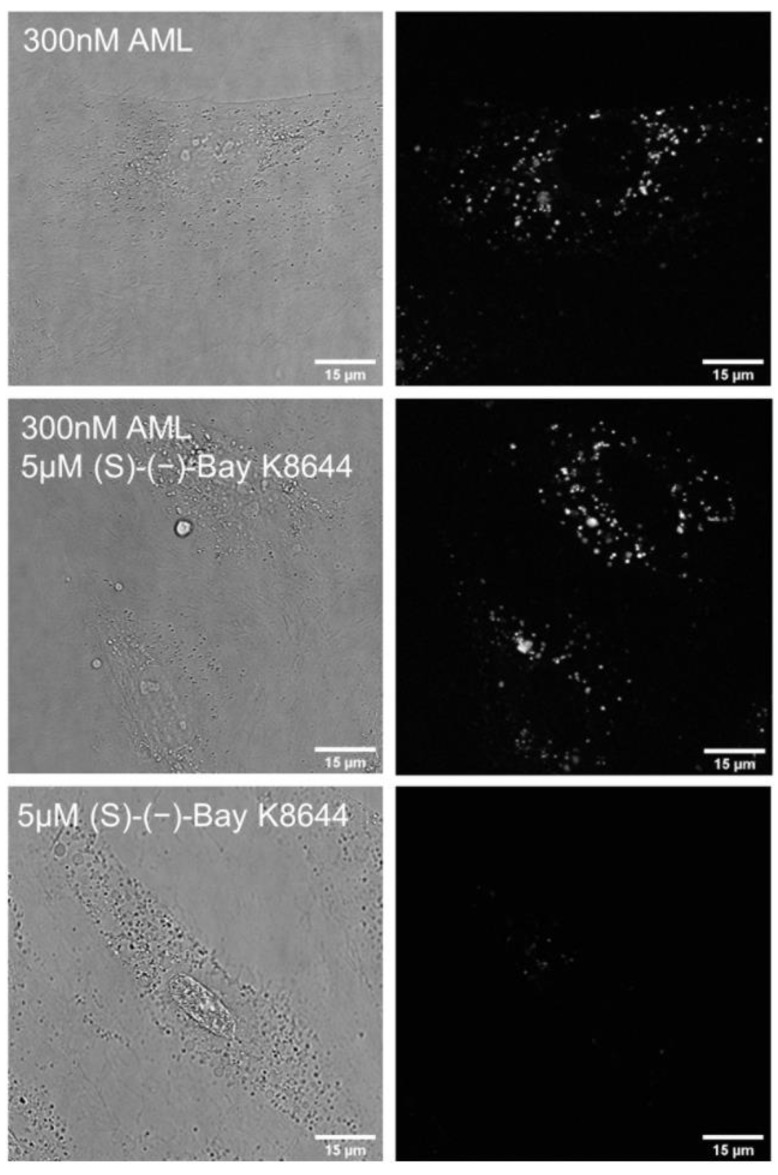
Phase contrast and DAPI channel fluorescence of living A7r5 cells. 300 nM AML (upper row), 5 µM (*S*)-(−)-Bay K8644 (middle) and competitive 300 nM AML + 5 µM (*S*)-(−)-Bay K8644. The cells were incubated for 1 h in HBSS/HEPES (pH 7.4) containing indicated concentrations of ligands, washed with HBSS and imaged in HBSS on confocal Leica SP8 microscope using settings as described in the Materials and Methods section.

**Figure 4 molecules-26-02997-f004:**
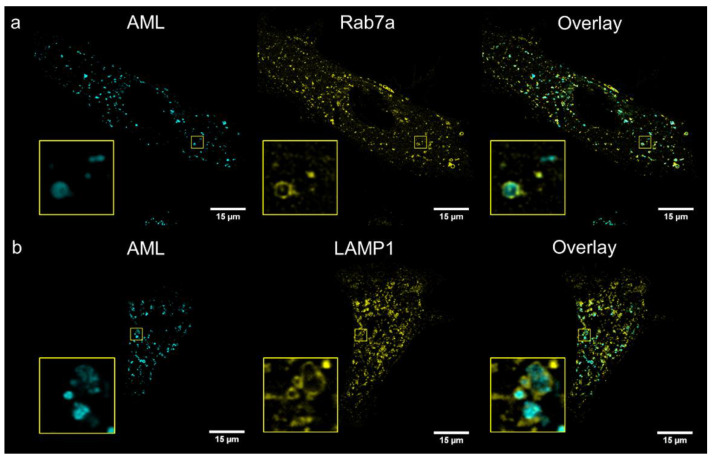
Confocal images of living A7r5 cells after 1 h incubation with 300 nM AML (cyan) and co-stained with CellLight^®^ emGFP trackers (yellow) (**a**) late endosomes marker Rab7a, (**b**) lysosomes marker LAMP1. Whole cells and selected ROIs 5 × 5 µm are shown. The cells were incubated in HBSS supplemented with 20 mM HEPES (pH 7.4), then washed twice and imaged on confocal Leica SP8 microscope as described in the Materials and Methods section.

**Figure 5 molecules-26-02997-f005:**
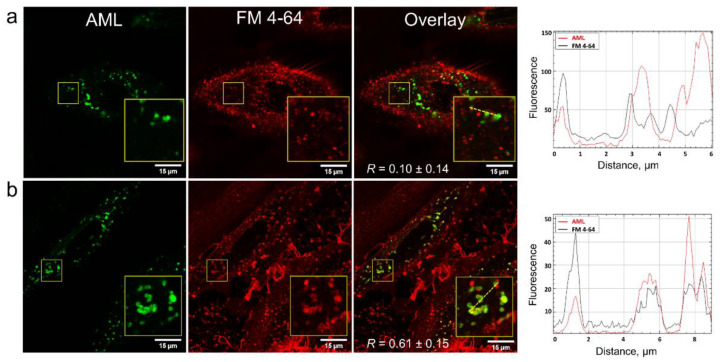
Confocal images showing live A7r5 cells counterstained with AML (green) and FM 4-64 (red). Cells were incubated for 15 min at 37 °C in HBSS/HEPES containing 300 nM AML and 5 mM FM 4-64 dye and imaged in HBSS in confocal SP8 microscope immediately after (**a**) or after 75 min (**b**). Whole cells and selected ROIs 12 × 12 µm are shown. Pearson’s co-localization coefficients (*R*) mean ± S.D. were determined using ImageJ 1.53c software from seven ROIs 2 × 2 µm containing AML signal. The line profiles were taken across the yellow dashed lines (AML—red, FM 4-64—black) and calculated using ImageJ 1.53c software.

**Figure 6 molecules-26-02997-f006:**
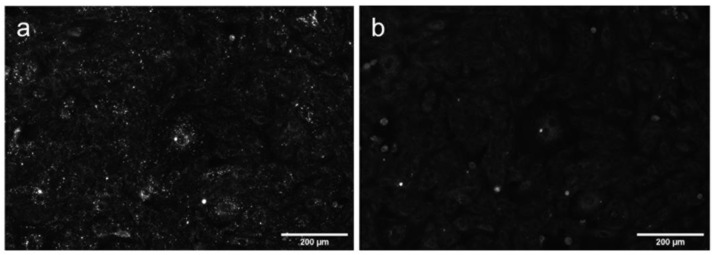
DAPI channel fluorescence of living A7r5 cells treated with AML and NH_4_Cl. The cells were incubated (**a**) with 300 nM AML for 15 min at 37 °C in HBSS/HEPES washed 2 times with HBSS; (**b**) the same as in (**a**), then treated with 10 mM NH_4_Cl in HBSS/HEPES, immediate reading. The images were acquired on wide-field Lionheart FX automated microscope using settings as described in the Materials and Methods section.

**Figure 7 molecules-26-02997-f007:**
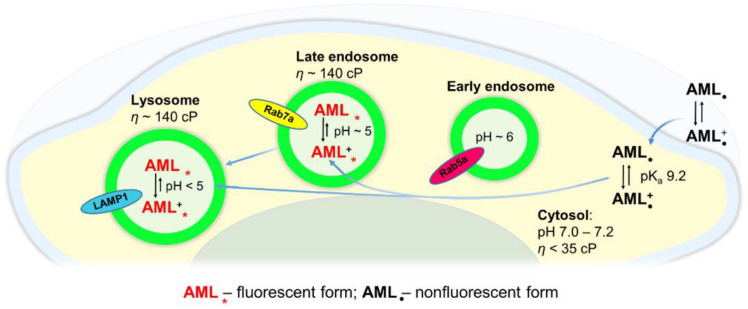
Proposed mechanism of AML interactions with cellular vesicles. AML penetrates the plasma membrane and accumulates in acidic compartments giving the fluorescence enhancement in the viscous medium. Protonated AML molecules are trapped in the late endo-/lysosomal vesicles.

**Table 1 molecules-26-02997-t001:** Photophysical properties of AML in different solutions.

	PBS	PBS + 0.1% SDS	PBS + 10% BSA	PBS + 10% HSA	Glycerol
λ_abs_ (nm)	365	365	365	365	365
λ_em_ (nm)	450	450	430; 445	430; 445	450
*Φ* (%)	3.5	4.0	53 for AML 55 for (*S*)-AML	47 for AML 44 for (*S*)-AML	40
*ε* (M^−1^ × cm^−1^)	6600	6700	7100	7200	8000
*τ* (ns)	1.0	1.2	6.4	7.9	5.6
*τ_rad_* (ns)	28.5	30.0	11.6	16.8	14.0

## Data Availability

Not applicable.

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
