# Peer review of "Direct Visualization of Amlodipine Intervention into Living Cells by Means of Fluorescence Microscopy"

_molecules, 2021, doi:10.3390/molecules26102997_

Round 1

Reviewer 1 Report

The manuscript by Quentin et al. explores and extensively characterizes the fluorescent properties of Amlodipine (AML). This compound is an L-type calcium channels antagonist used in patients with cardiac conditions such as angina. The authors performed in vitro experiments with direct visualization of AML in cardiac cell lines and describe the distribution of amlodipine inside the cells. Based on the physicochemical properties of AML and its distribution in live cells, the authors propose a mechanism of AML interactions with different cell compartments. 

The paper is very well written. The introduction enunciates the problem, the results are extensive and well presented, and the methods described in detail. 

Some minor suggestions/comments to the manuscript are:
- lines 189-192 - it is possible that the interaction AML-Cav1.2 changes AML conformation resulting in lower fluorescence emission. While this might be implicit in the sentence, it could be included further explicitly.
- Supplementary figures would benefit from a general conclusion in the figures legends of the significant findings depicted in the figure. While this is stated in the main text for most figures (see, for example, figure S8), this would benefit the manuscript's readability.

Author Response

Response to Reviewer 1 Comments:

Point 1: - lines 189-192 - it is possible that the interaction AML-Cav1.2 changes AML conformation resulting in lower fluorescence emission. While this might be implicit in the sentence, it could be included further explicitly.

Response 1: We thank the reviewer for this remark and have changed the text accordingly.

“There are two possibilities why we could not observe staining of CaV1.2 by AML: either interaction with the channel does not lead to fluorescence increase, presumably because AML takes the molecular conformation with lower fluorescence emission than in a stabilized form in BSA/HSA solutions, or the density of receptors is too low for the detection and is masked by receptor-independent staining.”

Point 2: - Supplementary figures would benefit from a general conclusion in the figures legends of the significant findings depicted in the figure. While this is stated in the main text for most figures (see, for example, figure S8), this would benefit the manuscript's readability.

Response 2: We appreciate this advice and have updated Supplementary figures legends accordingly.

Reviewer 2 Report

The authors in this article show the direct visualization of Amlodipine in live cells. They further assessed the mechanism of fluorescence quenching in aqueous solutions and examined the impact of pH, polarity and viscosity of the environment as well as protein binding on the spectral properties of Amlodipine in vitro. This article would be certainly in the interest of the readers with progress in understanding fluorescence properties of Amlodipine. The following minor concerns should be addressed.

  1. Author should mention proper reasons that why the viscosity is higher at lower pH?

  1. DMSO is polar aprotic and not as viscous as glycerol, then why does the fluorescence intensity should increase? Is there any charge transfer reaction going on?

  1. In LFP Figure S6, this is not clear that why is delta O.D. increasing with respect to time for Fig S6a, unlike Fig 6b? Moreover, what is the reason behind the evolution of a new peak around 430nm for AMD solution in PBS containing 10% BSA. Are there any possibilities of excited-state electron/proton transfer between AMD and BSA/HSA?

  1. In Figure 1(d, e) scale bar 15uM and ROI 15uM square are not properly scaled.

  1. Fluorescence anisotropy experiment will be indicative of the restriction of AMD tumbling in viscous solution.

Author Response

Response to Reviewer 2 Comments

Point 1: Author should mention proper reasons that why the viscosity is higher at lower pH?

Response 1: We thank the reviewer for this question. It is well known, that lysosome functions and composition depend on their acidic pH, and accumulation of various biomolecules leads to higher viscosity of lysosomes compared to cytosol. In our study, we acutely increased lysosomal pH by adding NH4Cl to the culture medium and saw accompanying decrease in Amlodipine fluorescence. Water viscosity is not influenced by pH, but lysosomes contain complex mixture of biomolecules, which might be affected by hydrogen ion concentration, thus leading to decreased lysosome viscosity. However, a more likely explanation is that increasing lysosomal pH releases the trapped amlodipine either because of increased membrane permeability or because of loss of protonation and diffusion through the membrane. Given these explanations, we did not address the effect of NH4Cl treatment on lysosome viscosity.

Point 2: DMSO is polar aprotic and not as viscous as glycerol, then why does the fluorescence intensity should increase? Is there any charge transfer reaction going on?

Response 2: We are grateful for this question. Our current quantum chemical calculations didn’t show that the AML fluorescence in DMSO is related to any charge transfer. The red shift of the fluorescence emission maximum might be because of the dimers or more complex aggregates formation. The more detailed investigations to clarify this issue will be conducted in the nearest future and published separately because they require significantly more complex calculations.

Point 3: In LFP Figure S6, this is not clear that why is delta O.D. increasing with respect to time for Fig S6a, unlike Fig 6b? Moreover, what is the reason behind the evolution of a new peak around 430nm for AMD solution in PBS containing 10% BSA. Are there any possibilities of excited-state electron/proton transfer between AMD and BSA/HSA?

 Response 3: We a sorry for a typo, the curve numbers in Fig. S6b were misplaced. The correct order is 1 (black), 2 (red), 3 (blue). In this case, there is no contradiction between Figs. S6a and S6b and the kinetic curves. Actually, no electron or proton transfer between AML and BSA/HSA was observed, because the delayed fluorescence spectrum of AML (recorded after 200 µs) is exactly the same as its prompt fluorescence spectrum.

Point 4: In Figure 1(d, e) scale bar 15uM and ROI 15uM square are not properly scaled.

 Response 4: We thank the reviewer for spotting it out.  The Figure 1 (d,e) was corrected accordingly.

Point 5: Fluorescence anisotropy experiment will be indicative of the restriction of AMD tumbling in viscous solution.

Response 5: We thank for this suggestion. Indeed, we are currently starting study of extensive characterization of newly developed fluorescent probes based on AML. We agree that these results are very useful for scientific community and hope to present them in a separate study in the nearest future.